# Experimental investigation of ant traffic under crowded conditions

Laure-Anne Poissonnier[1†], Sebastien Motsch[2†], Jacques Gautrais[1], Camille Buhl[3], Audrey Dussutour[1]*

[1]Research Center on Animal Cognition (CRCA), Center for Integrative Biology (CBI), Toulouse University, CNRS, UPS, 31062 Toulouse, France; [2]Arizona State University, Tempe, United States; [3]School of Agriculture, Food and Wine, The University of Adelaide, Adelaide, Australia

**Abstract** Efficient transportation is crucial for urban mobility, cell function and the survival of animal groups. From humans driving on the highway, to ants running on a trail, the main challenge faced by all collective systems is how to prevent traffic jams in crowded environments. Here, we show that ants, despite their behavioral simplicity, have managed the tour de force of avoiding the formation of traffic jams at high density. At the macroscopic level, we demonstrated that ant traffic is best described by a two-phase flow function. At low densities there is a clear linear relationship between ant density and the flow, while at large density, the flow remains constant and no congestion occurs. From a microscopic perspective, the individual tracking of ants under varying densities revealed that ants adjust their speed and avoid time consuming interactions at large densities. Our results point to strategies by which ant colonies solve the main challenge of transportation by self-regulating their behavior.
DOI: https://doi.org/10.7554/eLife.48945.001

*For correspondence:
dussutou@gmail.com

†These authors contributed equally to this work

Competing interests: The authors declare that no competing interests exist.

## Introduction

Many organisms such as herds of migrating wildebeests, swarms of insects and bacteria, starling flocks, fish shoals or pedestrian crowds take part in flow-like collective movements (*Ball, 2004*; *Berdahl et al., 2013*; *Berdahl et al., 2018*; *Buhl et al., 2006*; *Chowdhury et al., 2005*; *Fourcassié et al., 2010*; *Giardina, 2008*; *John et al., 2009*; *Moussaïd et al., 2011*; *Sumpter, 2010*; *Vicsek and Zafeiris, 2012*; *Zhou et al., 2008*). In most cases, all individuals cruise along the same path in a unique direction, which facilitates coordination. The task of maintaining a smooth and efficient movement becomes more challenging when individuals travel in opposite directions and are bound to collide (*Fourcassié et al., 2010*). Along with humans (*Helbing et al., 2005*; *Moussaïd et al., 2011*), ants are one of the rare animals in which collective movements are bidirectional. Ants are central-place foragers, which entails a succession of journeys between their nest and their foraging site. When exploiting large food sources, many species lay chemical trails along which individuals commute back and forth (*Czaczkes et al., 2015*; *Gordon, 2014*). The flow of individuals on these trails can reach several hundred ants per minute (*Couzin and Franks, 2003*). Yet, ants seem to fare better than us when it comes to traffic management (*Burd et al., 2002*; *Dussutour et al., 2004*; *Fourcassié et al., 2010*; *Hönicke et al., 2015*). However, we lack direct experimental evidence showing that ants do not get stuck in traffic jam at high density (*Burd et al., 2002*; *Hönicke et al., 2015*; *John et al., 2009*).

In traffic engineering, the relation between the density of individuals $k$ and the flow $q$ (the speed $v$ times the density $k$) is often described via the fundamental diagrams (*Chowdhury, 2000*; *Greenshields et al., 1935*; *Helbing, 2009*; *Helbing, 2001*; *Helbing and Huberman, 1998*; *Hoogendoorn and Bovy, 2001*; *Pipes, 1953*; *Underwood, 1960*) (*Figure 1A*). The speed-density

**eLife digest** Humans and ants are among the few species that engage in two-way traffic. Maintaining a smooth and efficient traffic flow while avoiding collisions is challenging for humans. Yet ants seem to be masters of traffic management. They can efficiently move back and forth between their nests and food without overtaking or passing each other, forming a steady stream of traffic. Few studies have looked at how ants maintain such a smooth flow even as the number of ants on a path increases.

Now, Poissonnier, Motsch et al. have designed an experiment to investigate whether ants can maintain their steady stream of traffic when their path to food gets more crowded. This involved manipulating the density of ants using a combination of different sized colonies (ranging from 400 to 25,600 Argentine ants) and changing the width of the bridge connecting the ants to their source of food. The experiment was repeated 170 times, and data was collected on traffic flow, speed of the ants, and number of collisions.

For pedestrians and car traffic, the flow of movement will slow down if occupancy levels reach over 40%. Whereas in ants, the flow of traffic showed no signs of declining even when bridge occupancy reached 80%. The experiments revealed that ants do this by adjusting their behavior to their circumstances. They speed up at intermediate densities, avoid collisions at large densities, and avoid entering overcrowded trails.

Studying ant traffic management has been a source of inspiration for scientists working with large groups of interacting particles in many fields. This includes molecular biology, statistical physics, and telecommunications. It may also have relevance for managing human traffic, particularly as scientists develop autonomous vehicles that might be programmed to work together cooperatively as ants do.

DOI: https://doi.org/10.7554/eLife.48945.002

and flow-density diagrams vary depending on the system under scrutiny but share similar properties. First, the flow $q$ increases with the density $k$ from zero to a maximum value and then decays until it goes back to zero at the so-called maximum jam density $k_j$. The flow-density curves are usually concave with an optimal value for $k$ on the path at which maximum flow or capacity is reached (*Helbing and Huberman, 1998*). Second, the speed is maximum when an individual is traveling alone (free flow speed $v_f$) and decreases with the density $k$. The speed becomes zero and individuals stop at jam density that is $v(k_j)=0$ (*Nagatani, 2002*; *Treiber et al., 2000*).

Despite their apparent efficiency in traffic management, few studies have investigated the relation between density, speed and flow in ants (*Burd et al., 2002*; *Gravish et al., 2015*; *Hönicke et al., 2015*; *John et al., 2009*). In leaf-cutting ants (*Burd et al., 2002*) and fire ants (*Gravish et al., 2015*) speed decreases when density increases while for wood ants (*Hönicke et al., 2015*) and mass raiding ants (*John et al., 2009*) the speed remains constant when density increases. However, the range of densities tested was large enough to observe traffic jams only in the study conducted with fire ants traveling in tunnels (*Gravish et al., 2015*). The highest densities as well as the estimated occupancy (fraction of area covered by ants), recorded in leaf-cutting ants (*Burd et al., 2002*), wood ants (*Hönicke et al., 2015*) and mass raiding ants (*John et al., 2009*) were relatively low: 0.8 ants.cm$^{-2}$ (occupancy 0.20), 0.6 ants.cm$^{-2}$ (0.13) and 0.3 ants.cm$^{-2}$ (0.10) and not sufficiently high to generate a traffic jam as ants never exceeded the capacity of the trail, that is the maximum value of the flow allowed by the trail width.

Here, we investigated if ants succeed in maintaining a smooth traffic flow and avoid traffic congestion under the widest possible range of densities. We used the European supercolony of Argentine ants *Linepithema humile*, which is a major pest around the world and the largest recorded society of multicellular organisms (*Giraud et al., 2002*). In our experiment, a colony was connected to a food source using a bridge (*Figure 1B*). To manipulate density, we used a combination of bridges of different widths (5, 10 and 20 mm) and experimental colonies of different sizes (from 400 to 25,600 ants). We conducted a total of 170 experiments. The flow $q$ and the density $k$ were recorded on each experiment every second for one hour giving us 612,000 flow/density (non-

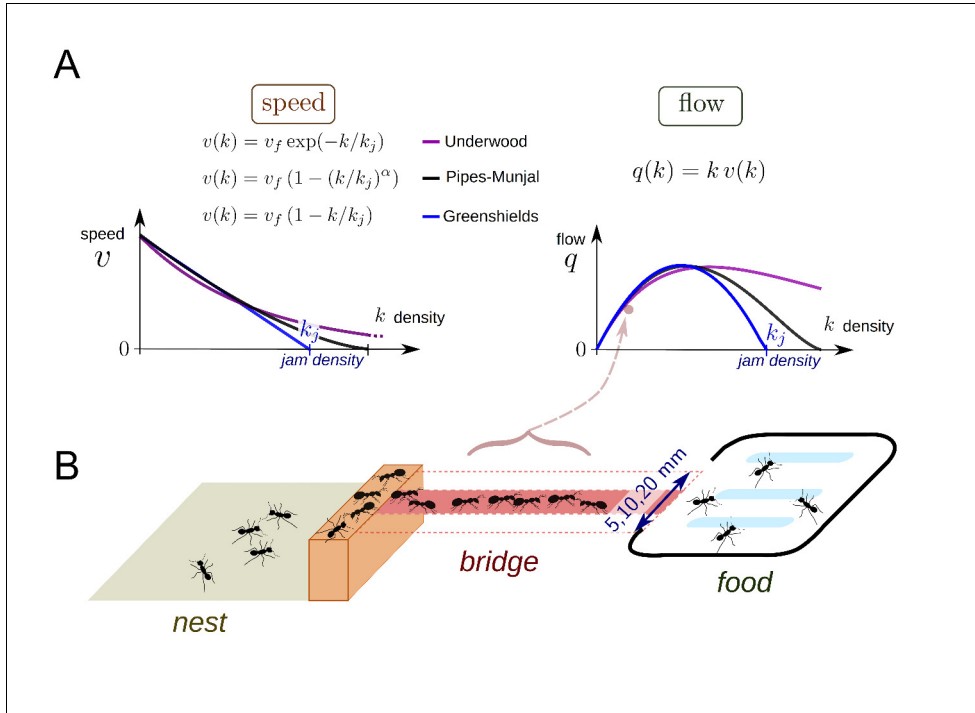

**Figure 1.** Theoretical fundamental diagrams and data collection. (**A**) Three functions are often used to describe vehicular, fluid or pedestrian flows. The Greenshields function (*Greenshields et al., 1935*) was the first speed-density relation function used to analyze data recorded for vehicular traffic in the field. This function assumes a linear relationship between $v$ and $k$ leading to a parabolic shape relation between $q$ and $k$. There are two parameters to be determined: the free speed $v_f$ corresponding to the speed of an individual without contact and the jam density $k_j$, which is a threshold over which $q$ becomes null. In the Pipes-Munjal function (*Pipes, 1953*), used for both pedestrians and car traffic, the relationship $v$ - $k$ is given by a power-law. This function requires a third parameter $\alpha$ to describe the function. The third function is the Underwood function (*Underwood, 1960*), which often describes well pedestrians or fluids traffic, where v decays exponentially rather than linearly. These functions made no a priori assumptions as to the behavior of the individuals and speed-density relationships are only obtained by fitting function to observed traffic data. (**B**) Experimental set-up. During an experiment, an ant colony (400 to 25,600 workers, 35 colonies in total) had access to a source of food (1M sucrose solution) placed at the end of a bridge (width: 5, 10 or 20 mm). The colonies were starved for five days before each experiment. The sucrose solution was spread over a large surface so that all ants had access to the food. The traffic on the bridge was recorded by a video camera for one hour. Inbound and outbound ants were counted over 1·sec intervals. Counting began as soon as the first ant crossed the middle of the bridge. A total of 170 experiments were performed.

DOI: https://doi.org/10.7554/eLife.48945.003

The following figure supplements are available for figure 1:

**Figure supplement 1.** Probability of feeding when an ant reached the food as a function of colony size.

DOI: https://doi.org/10.7554/eLife.48945.004

**Figure supplement 2.** Travel time without contacts as a function of bridge width.

DOI: https://doi.org/10.7554/eLife.48945.005

independent) observations pairs. We succeeded in generating large variations of density (from 0 to 18 ants.cm$^{-2}$) and occupancy (from 0 to 0.8).

## Results

Before pooling the data of all the 170 experiments to analyze the fundamental diagrams, we assessed the reliability of our experimental protocol with regard to foraging behavior. First, we verified that feeding behavior was not affected by the number of ants reaching the food (*Figure 1—figure supplement 1*). Most ants ate once at the food source, precluding the existence of a negative

feedback caused by crowding at the feeding site that could affect food exploitation and recruitment behavior (*Grüter et al., 2012*). Second, we controlled that the bridge width itself did not affect ant speed. In absence of interactions and when ants were traveling alone, their speed was similar regardless of the bridge width (*Figure 1—figure supplement 2*).

## Macroscopic traffic dynamics in ants

We first studied ant traffic at a macroscopic level. The flow of ants $q$ heading in both directions was plotted as a function of density in *Figure 2A*. The flow $q$ increased with the density $k$ to a certain point and then it remained constant. We analyzed the relationship between $k$ and $q$ using three different macroscopic traffic functions (*Greenshields et al., 1935*; *Pipes, 1953*; *Underwood, 1960*) (*Figure 2B*). All the parameters of the functions were fitted using a nonlinear least squares fit procedure (*Figure 2B*, *Supplementary file 1*). All the statistical models performed similarly well but failed at predicting the data at intermediate and large densities. Thus, based on experimental data, we introduced a two-phase flow function to describe the relationship $q$-$k$ as a piecewise linear function, with first a linear increase of the flow, followed by a constant value when the jamming density was reached.

$$\text{Two}-\text{phase flow function}: q(k) = k \cdot v \text{ if } k \leq k_j \text{ and } q(k) = k_j \cdot v = q_j \text{ if } k > k_j \tag{1}$$

Then, we conducted a model selection analysis using Akaike weights to assign conditional probabilities to all statistical models. Thanks to our large dataset, the result was unequivocal: the two-phase statistical model was selected (*Figure 2C*, *Figure 2—figure supplement 1*, *Supplementary file 1*).

Why did the ants not jam or clog, as one would expect in usual traffic situations? This could result from a spatiotemporal organization of the flow at high density (*Fourcassié et al., 2010*). The traffic is considered as spatially organized when the flows of inbound and outbound ants are not completely intermingled and lane segregation occurs (*Couzin and Franks, 2003*; *Fourcassié et al., 2010*). Temporal organization arises when oscillatory changes in the flow direction are observed and the traffic becomes intermittently unidirectional that is alternating clusters of inbound and outbound ants occurs (*Fourcassié et al., 2010*). Both type of organizations limits the rate of time-consuming contacts (collisions) and allows ants to maintain a smooth traffic (*Fourcassié et al., 2010*). However, in our experiments, no clear evidence of such organizations was observed. In contrast, when ant density reached a critical threshold, inbound and outbound flows became intermingled temporally (*Figure 3—figure supplement 1*) and spatially (*Figure 3A*, *Figure 3—figure supplement 2*, *Video 1* and *Video 2*). In addition, contrary to pedestrian traffic (*Helbing et al., 2005*; *Moussaïd et al., 2011*), the relationship between density $k$ and flow $q$ was only marginally influenced by the degree of asymmetry in the flows (*Figure 3B*, *Figure 3—figure supplement 3*). Simply put, it did not increase faster with the density $k$ when traffic was mostly unidirectional (*i.e.* proportion of outbound

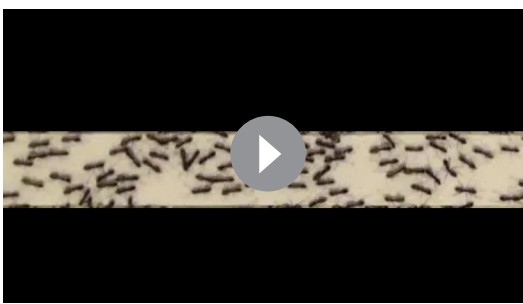

**Video 1.** Ants traveling on a 20 mm wide bridge. The video shows ants traveling between their nest and a food source 10·min after the beginning of the experiment, when the outbound and nestbound flows of ants were at equilibrium.
DOI: https://doi.org/10.7554/eLife.48945.012

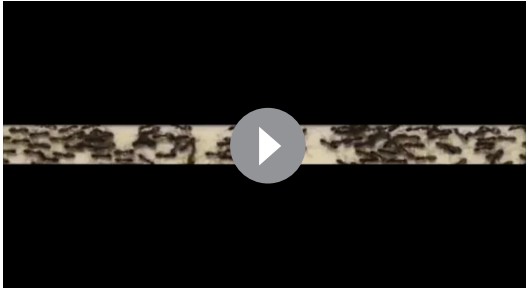

**Video 2.** Ants traveling on a 5 mm wide bridge. The video shows ants traveling between their nest and a food source 10·min after the beginning of the experiment, when the outbound and nestbound flows of ants were at equilibrium.
DOI: https://doi.org/10.7554/eLife.48945.013

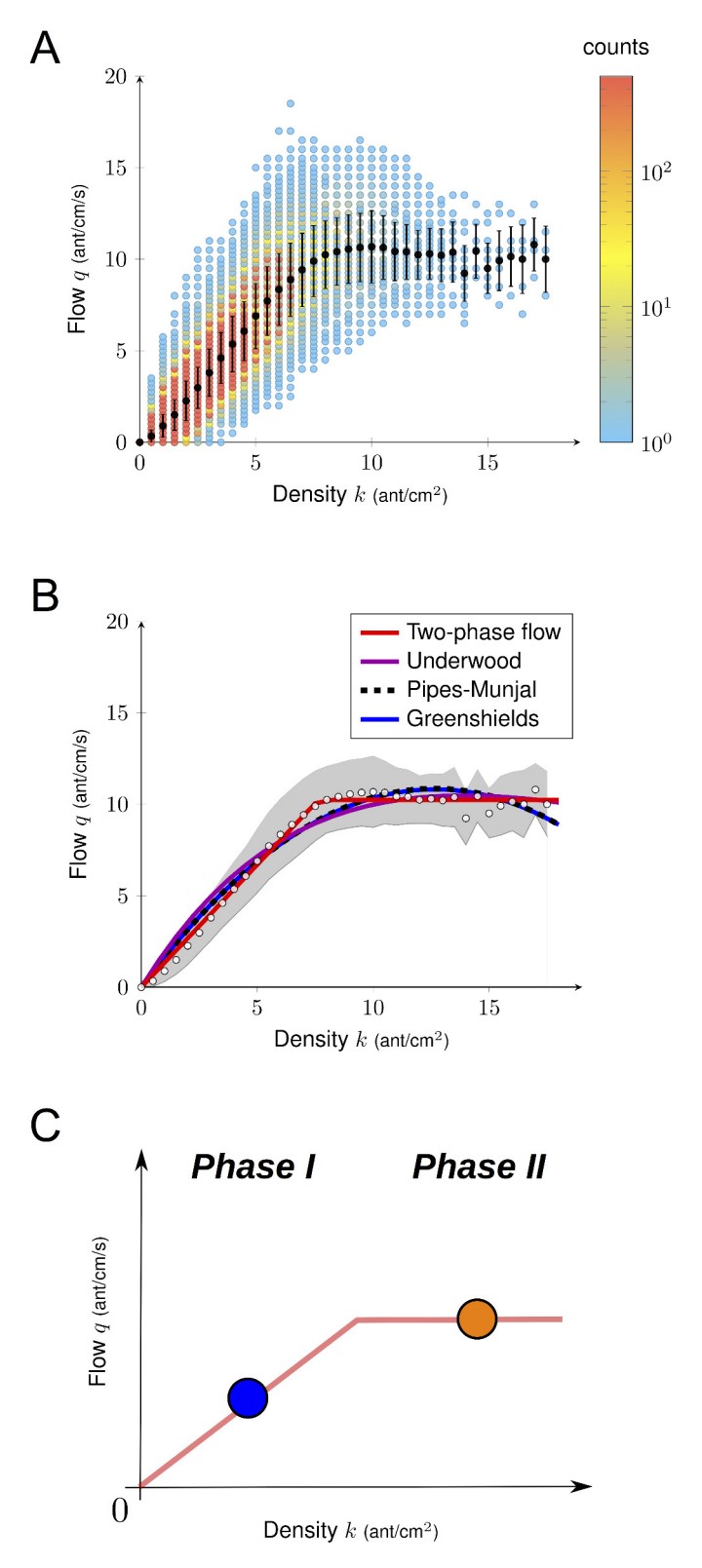

**Figure 2.** Empirical fundamental diagram. (**A**) Density flow relationship. Both the density and the flow were measured over a time interval of one second and each point in the diagram corresponds to one data point (N = 612000 in total, N = 10303 for $k > 8$). Black dots indicate the mean and bars are standard deviations. (**B**) Non-linear fitting of the fundamental diagrams for the four functions tested (see *Supplementary file 1*). White dots represent the experimental mean flow and the gray region its standard deviation. (**C**) Illustration of a two phase flow diagram. At low density, the flow

*Figure 2 continued on next page*

*Figure 2 continued*

increases linearly with the density, whereas at large density the flow saturates and remains constant. Phase 1 ($k < 8$): $q$ increases linearly with $k$, ants move freely. Phase 2 ($k > 8$): $q$ stops increasing with $k$, but does not decay.

DOI: https://doi.org/10.7554/eLife.48945.006

The following figure supplement is available for figure 2:

**Figure supplement 1.** Non-linear fitting of the fundamental diagrams without weights.

DOI: https://doi.org/10.7554/eLife.48945.007

flow $q_o$ close to 1 or 0, traffic considered as asymmetric) than when it was entirely bidirectional (*i.e.* proportion of outbound flow $q_o$ close to 0.5, traffic considered as symmetric). This is illustrated in *Figure 3B* by the tendency for the isoclines to run parallel to the y-axis (*Figure 3B*). Therefore, we focused our next analyses on individual behavior to understand how ants maintained a constant flow despite the increasing density.

## Microscopic traffic dynamics in ants

From an individual behavior perspective, most traffic-flow functions (*Greenshields et al., 1935*; *Pipes, 1953*; *Underwood, 1960*) suggest that individual speed decreases non-linearly with the density due to friction between individuals. However, the two-phase flow traffic function suggested that there was no evidence of such friction between ants when the density was below eight ants. $cm^{-2}$ that is the flow increased linearly, whereas above eight ants.$cm^{-2}$ frictions appeared but increased only linearly with the density that is the flow remained constant over a wide range of densities. How can we quantify such friction at the individual level?

A key factor determining ant speed is the number of contacts (*i.e.* collisions) experienced with nestmates, which makes ants stop and thus decays their overall speed (*Burd and Aranwela, 2003*; *Dussutour et al., 2005*; *Gravish et al., 2015*; *Wang et al., 2018*). To test if the number of contacts played the role of a hidden variable linking density and speed, we measured the number of contacts $C$, the density $k$ and the traveling time $T$ for a sample of 7900 ants individually tracked on a 2 cm section of the bridge. As density $k$ grew, the number of contacts $C$ increased linearly ($C = 0.61\,k$ *Figure 4A*) that is the larger the density, the higher the number of contacts. We observed a linear effect of the number of contacts $C$ on traveling time $T$, that is each contact slowed down the ants. ($T = T_0 + C \cdot \Delta T$ with $T_0 = 0.95$ s and $\Delta T = 0.24$ s, *Figure 4B*). $T_0$ can be interpreted as the traveling time to cross the bridge without contacts (free flow traveling time), whereas $\Delta T$ is the time lost per contact.

So far, density $k$ only had a negative impact on the speed $v$: it increased the number of contacts $C$ which in return increased traveling time $T$. However, in our two-phase diagram, density $k$ had no or minor effect on speed $v$ in the phase 1. Therefore, there had to be a positive effect of density $k$ on speed $v$ in this regime. Thus, the interplay between $T$, $k$ and $C$ had to be more subtle. To combine multiple effects, we estimated the expected traveling time $T$ depending on both density $k$ and number of contacts $C$. For a given number of contacts $C$, we visualized the average traveling time $T$ (see *Figure 5A*) for various values of density $k$, using local regression fitting. The vertical distance between two neighbors' curves was given by $\Delta T$. As expected, traveling time $T$ increased with the number of contacts $C$, but the key information is that density $k$ actually made traveling time $T$ decay initially (up to $k \approx 5$). For a given number of contacts $C$, ants moved faster on the bridge at intermediate density $k$ (*i.e.* $k \approx 5$). To further visualize this positive effect of the density, we estimated the free flow speed $v_f$, that is speed without contact $v_f = L/(T - C \cdot \Delta T)$, where $L = 2$cm is the length of the recording section on the bridge (*Figure 5B*). $v_f$ is first increasing with density $k$ up to $k \approx 5$ ants.$cm^{-2}$ and then decays back to its initial value. This phenomenon might be explained by a pheromone effect. It is well known that Argentine ants deposit pheromones both when leaving and when returning to the nest and travel faster on a well-marked trail (*Deneubourg et al., 1990*).

To incorporate both effects of the density, we proposed the following formula for the speed:

$$v(k) = \frac{L}{T_0 + \Delta T \cdot C(k)} \cdot \left( \alpha + \beta \cdot k \cdot e^{-\gamma \cdot k} \right) \tag{2}$$

where $C(k) = 0.61 \cdot k$ is the average number of collisions, the three other parameters $\alpha$, $\beta$ and $\gamma$ model

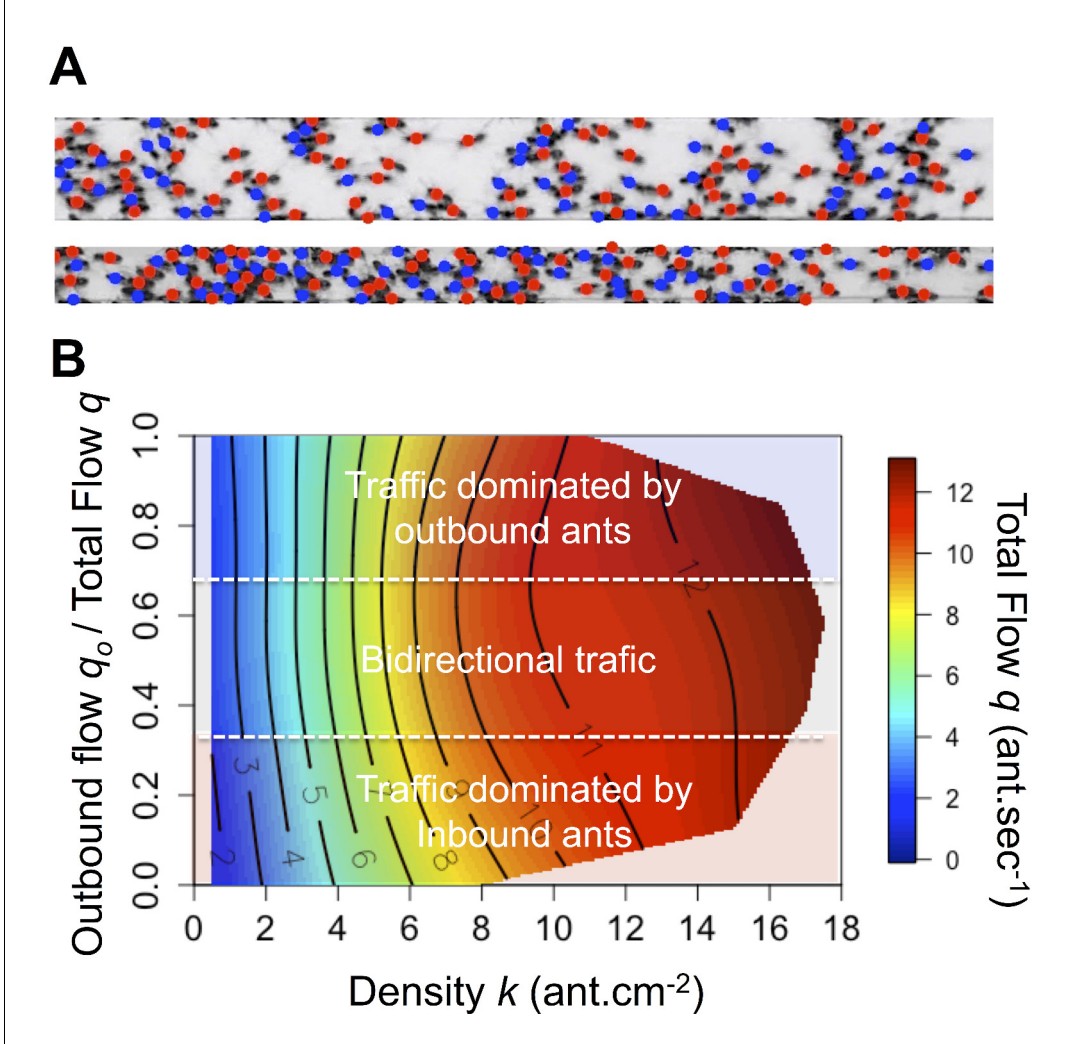

**Figure 3.** Spatial organization and flow asymmetry. (**A**) Illustration of the ants' spatial distribution on the 10 mm and 5 mm bridge. Colored dots correspond to travel direction. (**B**) Flow $q$ (color scale) as a function of density $k$ (x-axis) and asymmetry in the flows (y-axis). Traffic asymmetry was computed as the number of ants going to the food source per second (outbound flow) divided by the total number of ants traveling on the bridge (total flow $q$). When the proportion of outbound flow is 1 or 0 the traffic is unidirectional that is asymmetric, while if it is close to 0.5 the traffic is bidirectional that is symmetric. Red indicates the highest values for $q$ on the response surface, with values descending to lowest values in dark blue regions. Response surfaces were visualized using non-parametric thin-plate splines, which were fitted using the *fields* package (*Furrer et al., 2009*) in the statistical software R version 3.5.0. The response surface regression analyses yielded significant relationships as follows: $R^2 = 0.82$, $p<0.001$. The density is the main parameter affecting the flow while traffic asymmetry has only a marginal effect. Main effects: standardized beta $\beta_{density} = 0.980$, $\beta_{density2} = -0.126$; Marginal effects: $\beta_{Asymmetry2} = -0.040$, $\beta_{Asymmetry} = 0.026$ and $\beta_{Density*Asymmetry} = 0.02$. Flow isoclines run parallel to the Y-axis indicating that the flow depends mainly on density and is independent of traffic asymmetry.
DOI: https://doi.org/10.7554/eLife.48945.008

The following figure supplements are available for figure 3:

**Figure supplement 1.** Temporal organization of the traffic.
DOI: https://doi.org/10.7554/eLife.48945.009

**Figure supplement 2.** Spatial organization of the traffic.
DOI: https://doi.org/10.7554/eLife.48945.010

**Figure supplement 3.** Flow $q$ (y-axis) as a function of the asymmetry in the flows (x-axis) for three range of density $k$.
DOI: https://doi.org/10.7554/eLife.48945.011

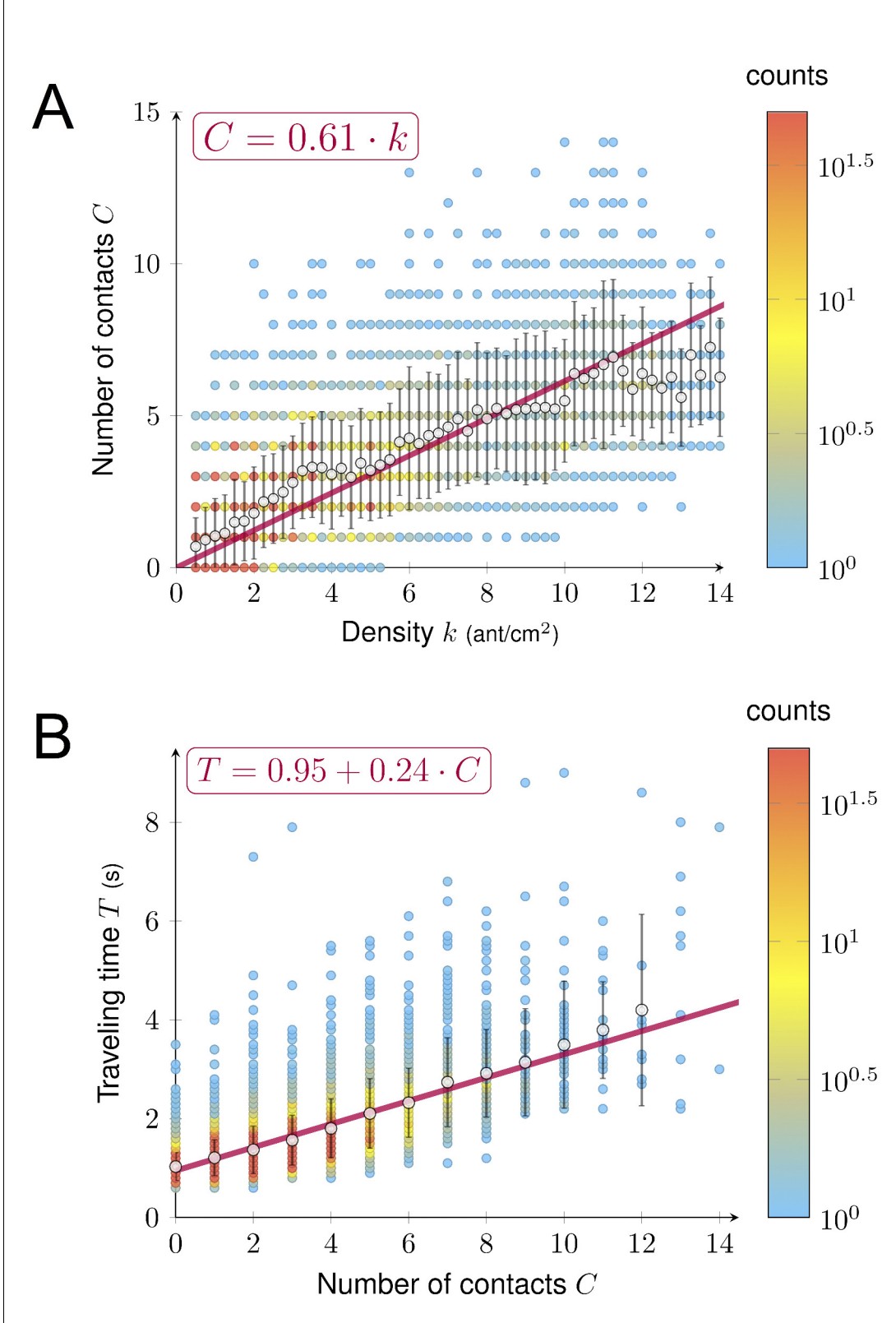

**Figure 4.** Relations between density, contact and traveling time. (**A**) Number of contacts $C$ an ant incurred during the crossing of the bridge depending on the density $k$. $C$ increased linearly with $k$ ($C = 0.61 \cdot k$, $R^2 = 0.77$). (**B**) Individual traveling time $T$ as a function of the number of contacts $C$. $T$ increased linearly with $C$ ($T = T_0 + C \cdot \Delta T$ with free traveling time $T_0 = 0.95$ s and time incurred by contact $\Delta T = 0.24$ s, $R^2 = 0.55$). We measured $C$ along with $k$ and the traveling time $T$ for a sample of 7900 ants individually tracked on a 2 cm section at the center of the bridge (98 to 364 ants followed for each

*Figure 4 continued on next page*

*Figure 4 continued*
direction, each experimental group size and each bridge width). Data were collected 10·min after the beginning of the experiment, when the outbound and nestbound flows of ants were at equilibrium. Data were issued from 42 experiments in total. White dots represent average. Error bars are standard deviations.
DOI: https://doi.org/10.7554/eLife.48945.014
The following figure supplement is available for figure 4:

**Figure supplement 1.** Time lost per contact as a function of collisions type.
DOI: https://doi.org/10.7554/eLife.48945.015

the pheromone effect: $\alpha$ corresponds to the intrinsic attractiveness of the unmarked bridge, $\beta$ represents the positive effect of $k$ and $\gamma$ measures the range where the pheromone effect might occur. These three parameters were estimated using a non-linear regression algorithm ($\alpha=0.812 \pm .009$, $\beta=0.160 \pm .010$, $\gamma=0.156 \pm .007$). We observed a decay of the speed $v$ when k increased (*Figure 6A*). From this estimation of the speed, we deduced the following formula for the flow:

$$q(k) = k \cdot v(k) \tag{3}$$

We plotted in *Figure 6B* the predicted flow $q$ and the experimental observations of the sub-sample dataset (N = 7900 observations) used for the estimation. We noticed a clear agreement between the data and the model predictions. The plateau reached was $q \approx 10$ ants.cm$^{-1}$.s$^{-1}$. Thus, even though increasing density $k$ generated more contacts impacting negatively the flow $q$ through their effect on traveling time $T$, when $k < 5$ ant.cm$^{-2}$ ants moved faster, which positively impacted the flow $q$. These two effects counterbalanced, leading to a linear increase of the flow $q$ with density $k$ (phase 1). For $k > 8$ ants.cm$^{-2}$, despite the crowdedness of the trail, ants maintained a constant flow $q$. The speed $v(k)$ continued to decay due to the increase of contacts, but this negative effect on the flow $q(k)$ was offset by the increase of $k$. We recovered this phenomenon from our model by estimating the limit flow $q$ as the density $k$ increases:

$$\lim_{k \to \infty} q(k) = \frac{L}{\Delta T \cdot 0.61} \cdot \alpha \approx 11.09 \tag{4}$$

In other words, the flow in *Figure 6B* would merely increase for larger values of density $k$. However, experimentally the flow would have to decay eventually as the ant occupancy on the bridge cannot increase indefinitely. Given that the unoccupied portion of the bridge decayed with the density, it is remarkable that the number of contacts increased only linearly with the density according to *Figure 4A* (or one could argue even sub-linearly for $k > 10$ ants.cm$^{-2}$). Interestingly, we also observed that ants restrained themselves from leaving the nest to prevent overcrowding as $k$ never exceeded 18 ant.cm$^{-2}$ even though we increased colony size and reduced bridge width (number of ants entering the bridge for the largest colonies $\pm$ CI95: 2.69 $\pm$ 0.04, 4.34 $\pm$ 0.03 and 5.05 $\pm$ 0.03 ants.sec$^{-1}$ for the 5, 10 and 20 mm bridges). Furthermore, U-turns were seldom once the ants were traveling on the bridge (probability of turning back once on the bridge = 0.01).

## Discussion

Traffic jams are ubiquitous in human society where individuals are pursuing their own personal objective (*Youn et al., 2008*). In contrast, ants share a common goal: the survival of the colony, thus they are expected to act cooperatively to optimize food return. In pedestrian and car traffic (*Banks, 1999*; *Helbing et al., 2005*), at occupancy levels above 0.4 the flow starts to decline, while in ants, the flow showed no sign of declining even when occupancy reached 0.8. Here, by investigating traffic dynamics on a large range of densities, we demonstrated that ants seem to be immune to traffic congestion.

The traffic of pedestrians or vehicles is often ruled by external constraints (enforced rules), time consuming interactions (avoidance behavior) and negative feedbacks (jamming). In contrast, in ants, traffic is governed by positive feed-backs (trail following and reinforcement) and interactions which are time consuming but often beneficial for the colony as they promotes information transfer (*Bouchebti et al., 2015*; *Burd and Aranwela, 2003*; *Farji-Brener et al., 2010*; *Fourcassié et al.,*

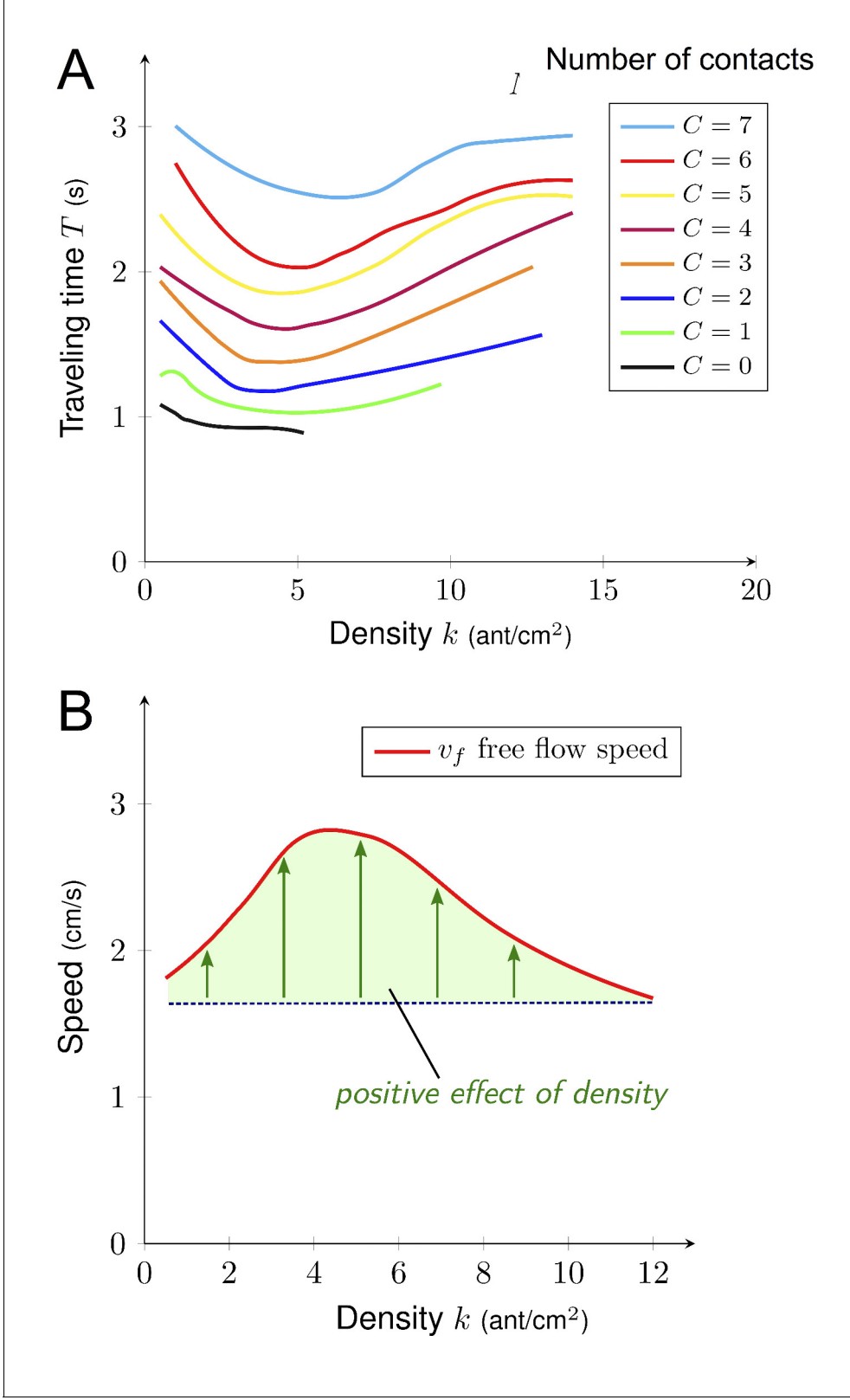

**Figure 5.** Free flow speed. (A) Traveling time $T$ depending on the density $k$ for a given number of contacts $C$. Higher $C$ induced longer $T$ and $T$ decreased until $k$ reached 6. (B) Free flow velocity $v_f$ as a function of density. Free flow velocity is computed as the distance (L) divided by free traveling time $T_0$ (traveling time without contact) that is $v_f = L/(T_0)$ where L (2 cm) is the length of the monitoring section on the bridge. Hence, $v_f = L/(T-C \cdot \Delta T)$ as $T_0 = T\,C \cdot \Delta T$ (**Figure 4B**) where $T$ is the individual traveling time, $C$ is the number of contacts and $\Delta T$ (0.24 s) is the time incurred by contact. Data were

*Figure 5 continued on next page*

*Figure 5 continued*

collected for a sample of 7900 ants individually tracked on a 2 cm section (L) at the center of the bridge (98 to 364 ants followed for each direction, each experimental group size and each bridge width). Data were collected 10 min after the beginning of the experiment, when the outbound and nestbound flows of ants were at equilibrium. Data were issued from 42 experiments in total. Each curve was obtained by local regression fitting with R command loess.

DOI: https://doi.org/10.7554/eLife.48945.016

*2010*; *Gordon, 2019*). Nonetheless, time spent interacting with other ants has strong consequences on traffic dynamics (*Gravish et al., 2015*). Short interaction time allows a smooth traffic over a wide range of densities whereas long interaction time promotes rapid slowing of traffic (*Gravish et al., 2015*). In fire ants, *Gravish et al. (2015)* found that interaction time was relatively short (0.45 s) and that the flow was relatively unaffected by density up to a certain critical value beyond which they observed the formation of traffic jams (*Gravish et al., 2015*). Here, we found that Argentine ants have an interaction time approximately twice shorter than fire ants, which could explain why we observed a smooth and efficient mixing of opposite flows over a broader range of densities.

The exact nature of the mechanisms used by Argentine ants to keep the traffic flowing in this study remains elusive, yet when density on the trail increases, ants seemed to be able to assess crowding locally, and adjusted their speed accordingly to avoid any interruption of traffic flow. Moreover, ants restrained themselves from entering a crowded path and ensured that the capacity of the bridge was never exceeded that is the maximum value of the flow allowed by the bridge width. Traffic regulation on trails ultimately allows Argentine ants to maintain a high rate of food return to the nest, an essential asset in the context of food competition occurring in natural environments. This balance between positive feedback and negative feedback appears to be generic in ants. For instance, in the nest-building context, behaviors such as 'individual idleness and retreating' at overcrowded digging sites supports an optimal accessibility of space facilitating excavation during nest construction in fire ants (*Aguilar et al., 2018*). Similarly, carpenter ants placed under threat and forced to escape through a narrow door distribute themselves uniformly over the space available instead of rushing toward the door and trampling on others (*Parisi et al., 2015*). As a result, contrary to humans (*Helbing et al., 2000*), they avoid clogging and evacuate efficiently (*Parisi et al., 2015*). Lastly, in garden ants, the interplay between trail following behavior and collisions allows ants to cope with bottleneck situations (*Dussutour et al., 2006*; *Dussutour et al., 2005*; *Dussutour et al., 2004*).

Overall, our results extend previous research on ant traffic organization (*Aguilar et al., 2018*; *Burd et al., 2002*; *Fourcassié et al., 2010*; *Gravish et al., 2015*; *Hönicke et al., 2015*; *John et al., 2009*) and show that ant prevented traffic jams from occurring and behaved as a self-organized biological adaptive system (*Camazine et al., 2003*; *Hemelrijk, 2005*). Similar phenomena of self-regulation ought to be found in other complex systems such as migrating animals (*Buhl et al., 2006*), cell machinery (*Leduc et al., 2012*), swarm of robots (*Aguilar et al., 2018*) and data traffic (*Valverde and Solé, 2004*) and could provide inspiration in all disciplines, ranging from molecular biology to automotive engineering. Although comparing ant traffic to human traffic might be a delicate task, as traffic in humans is neatly separated into unidirectional roadways, the rise of autonomous vehicles paves the way for new strategies to optimize traffic flow.

## Materials and methods

### Biological model

We used the Argentine ant *Linepithema humile*, an invasive species that uses mass recruitment through pheromone trails to exploit abundant food sources. Argentine ants have multiple queens and form supercolonies. Established supercolonies can contain up to $10^8$ workers (*Giraud et al., 2002*). Usually, foraging in Argentine ants is associated with high traffic between the nest and the food source in order to provide food for the whole colony. These ants are monomorphic (2.5 mm long). The ants were collected in Toulouse (France).

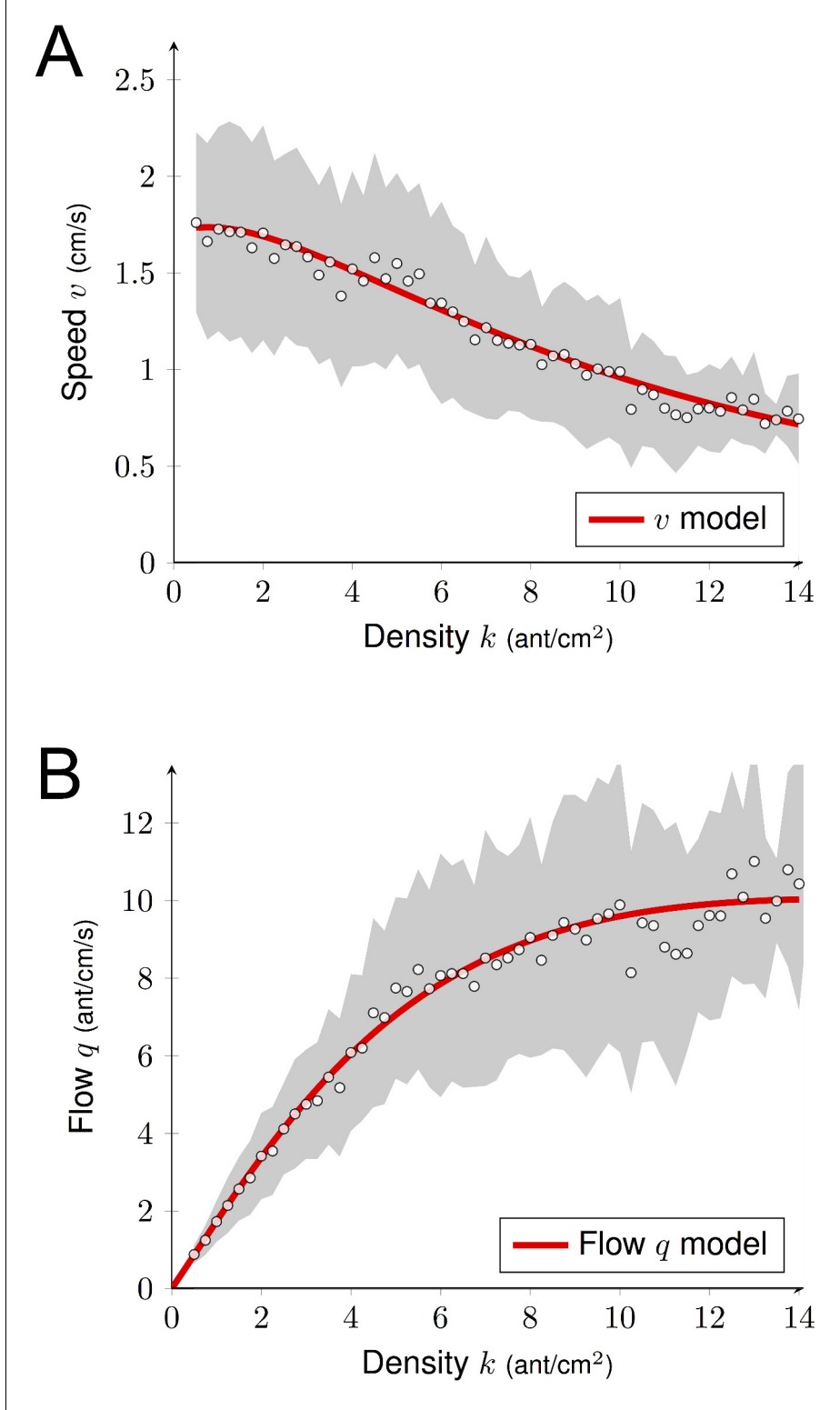

**Figure 6.** Predicted flow. (**A**) Estimation of the speed *v* as a function of the density *k*. We combined positive and negative effects of the density *k* on *v* using *Equation (2)*. Parameter estimation was done using non-linear regression method (R command *nls*). (**B**) Predicted flow *q* as a function of the density *k* using *Figure 4* and *Equation (3)*. We observed a linear growth of *q* (for *k* < 5–6), followed by a plateau (for *k* > 8). Experimental data correspond to the observations of a sub-sample dataset (7900 observations) from the full data set (612000 observations; *Figure 2A*) used to quantify the individual behavior. The gray shaded region represents the

*Figure 6 continued on next page*

*Figure 6 continued*

standard deviations. White dots are average computed from experimental data (N = 7,900). Data were collected 10 min after the beginning of the experiment, when the outbound and nestbound flows of ants were at equilibrium.

DOI: https://doi.org/10.7554/eLife.48945.017

## Experimental design

### Colonies

Ants were subdivided into 35 experimental colonies of different sizes by weighing the ants: 400, 800, 1600, 3200, 6400, 12800 and 25600 ants. A few thousands workers were kept into 'stock colonies' in order to maintain a stable number of ants in the experimental colonies throughout the duration of the experiment. The experimental colonies were installed in test tube nests placed in a rearing box (L x W x H: 25 × 10×9 cm for the small colony sizes 400, 800, 1600 and 3200 ants and 29 × 27.5×9 cm for the larger colony sizes 6400, 12800 and 25600) with walls coated with Fluon to prevent ants from escaping. The experimental colonies were kept at room temperature (25 ± 1°C) with a 12:12 L:D photoperiod. We supplied each experimental colony with water and a mixed diet of vitamin-enriched food twice a week (*Dussutour and Simpson, 2008*). Before each experiment, the experimental colonies were starved for five days, and the experiment started when the ants were given access to a food source placed on a platform (120 × 120 mm for small colonies - up to 3200, or 230 × 230 mm for the larger colonies) at the other end of a plastic bridge. The total length of the bridge was 170 mm and the bridge width was either 5 mm (narrow bridge), 10 mm (medium bridge) or 20 mm (large bridge). The food consisted of a 1M sucrose solution contained in small grooves (100 mm long for small colonies and 185 mm for large colonies) carved in a block of Plexiglas. To prevent crowding effects at the food source, the grooves were numerous enough (nine for small colonies, and 16 for large colonies) to give food access to a very large number of ants. The whole experimental set-up was isolated from any sources of disturbance by surrounding it with white paper walls. Throughout the experiments the traffic on the bridge was filmed from above for 60 min starting as soon as the first ant crossed the middle of the bridge. The number of replicates for each bridge width and each colony size ranged between 4 and 10 leading to a total of 170 experiments. Replicates on the same experimental colony were run at three weeks intervals. The temperature of the experimental room was 25°C.

## Data collection – Collective level

### Flow q

The flow represents the number of ants crossing a line per unit of time. The ant flow on the bridge was counted over a 1·sec period during 1·h for all the experiments. Counting began as soon as the first ant crossed a line drawn in the middle of the bridge. Ants seldom climbed on top of each other that is the flow remained two-dimensional in all experiments.

### Density K

The density represents the number of ants per unit of surface. The number of ants over a one $cm^2$ section encompassing the line drawn in the middle of the bridge was measured with ImageJ using the Analyze/Analyze Particles command, every half a second for one hour. Images were converted in binary images. Ants appeared black while the bridge appeared white. When the ant density was too high for the Analyze Particles command to discriminate the ants from one another, we divided the total area covered by black pixels by the mean area of a single ant. The mean area covered by a single ant on the bridge was measured on a total of 60 isolated ants on 10 experiments picked randomly. Ants covered on average 3.5 $mm^2$ excluding legs and antennae and 4.4 $mm^2$ with legs and antennae. For high densities ($k > 8$) we also analyzed visually a sample of 5000 pictures (*i.e.* by counting each ant) and compared it with automatically analyzed pictures. We found a good agreement although automatic analysis had a slight tendency to underestimate densities but only when density was higher than six ant.$cm^{-2}$ (difference between manual and automatic $-0.36$ ant.$cm^{-2}$+ /- IC95 0.02).

## Occupancy

For the sake of a comparison, we estimated occupancy (fraction of area covered by ants) obtained in previous studies (*Burd et al., 2002*; *Gravish et al., 2015*; *Hönicke et al., 2015*; *John et al., 2009*) based on density and ant size. We approximated the surface of an ant to a rectangle (body length x head width) and multiplied this area by 1.25 (ratio found in our experiment when dividing the area without legs and antennas by the area with legs and antennas) to include legs and antenna. We found that one ant cover a surface of 25 mm$^2$ in leaf-cutting ants (ant size: 8–12 mm, head width: 1.4–2.6 mm; *Burd et al., 2002*; *Nichols-Orians and Schultz, 1989*), 4.8 mm$^2$ in fire ants (ant size: 2.6–6.1 mm, mean 3.8 mm; head width: 0.6–1.4 mm; *Tschinkel, 2013*; *Tschinkel et al., 2003*), 22.25 mm$^2$ in wood ants (ant size: 7.7 mm, head width 2.3 mm; *Hönicke et al., 2015*), and 33.75 mm$^2$ in mass raiding ants (ant size: 18 mm, head width: 1.5 mm; *John et al., 2009*) giving us occupancy level of 0.20, 0.48, 0.13 and 0.10 corresponding to the densities: 0.8 ants.cm$^{-2}$, 10 ants.cm$^{-2}$, 0.6 ants. cm$^{-2}$ and 0.3 ants.cm$^{-2}$ respectively.

## Data collection – Individual level

### Experimental design assessment

To ensure that ants did not experience overcrowding at the food source which could affect the recruitment dynamic (*Grüter et al., 2012*), we measured the probability of feeding when an ant reached the food. In other words, we recorded if the ants fed before leaving the platform to return to the nest. This was done by following 4200 ants arriving at the food source (100 ants for two replicates for each experimental colony size and each bridge width).

To check that the ant traveling time was not affected by the experimental set-ups (*i.e.* the bridge width itself), we followed a total of 2133 ants that did not experience any collision while traveling on the bridge (706, 693 and 734 ants followed for the 5 mm, 10 mm and 20 mm bridges respectively on 34 different experiments; *Figure 1—figure supplement 2*).

### Traveling time *T* and number of contacts *C*

All individual behaviors were observed on a 20 mm section at the center of the bridge. The measurements began 10·min after the beginning of the experiment, when the outbound and nestbound flows of ants were at equilibrium. To test if the density affected the traveling time on the bridge, we recorded the travel duration *T* and the number of physical contacts *C* incurred while traveling. A contact was the result of either a head-on collision or a rear-end collision (when the head of an ant enters in contact with the gaster of the ant preceding it). Once the ant followed had crossed the 20 mm section of the bridge, we followed the next ant entering the section and so on. We followed 98 to 364 ants for each direction, each experimental group size and each bridge width, leading to total number of 7980 ants, of which 80 made a U-turn. Data were issued from 38 experiments in total. Head-on collisions and rear-on collisions were pooled together, as the time lost in each physical contact did not differ significantly (*Figure 4—figure supplement 1*). A physical contact lasted between 0.1 and 3.2·s (mean ± SD 0.18 ± 0.14 s). Automatic tracking was impossible due to high ant density, so all data were recorded semi-manually by two different persons using a homemade software AntEthoc-Combe-CRCA-CNRS (available upon request). We pressed various keys on a keyboard when an ant 1) entered or left the observation zone, 2) contacted another ant and 3) made a U-turn. Thus for each ant followed, the software gave us the travel time and the number of contacts.

## Statistical analysis

All the statistical analysis were done using R version R 3.5.0 (*Crawley, 2012*). All the parameters of the fundamental diagrams were fitted using a nonlinear least squares fit procedure (command nls; *Baty et al., 2015*). This procedure used Gauss-Newton algorithm to find the parameters that minimize the mean square error between the experimental data and the model prediction. The results of the estimations for the four functions are given in *Supplementary file 1*. A model selection using the Akaike weights (AW) has been conducted to assign a conditional probability for each statistical model. Source codes are available (*Poissonnier et al., 2019*).

## Data and materials availability

Data are available from the Dryad Digital Repository: https://doi.org/10.5061/dryad.8q58jg3 (*Poissonnier et al., 2019*).

## Acknowledgements

We are thankful to Rhonda Olson and Julie-Anne Popple for proofreading the article.

## Additional information

### Funding

| Funder | Grant reference number | Author |
|---|---|---|
| National Science Foundation | DMS-1515592 | Sebastien Motsch |
| Centre National de la Recherche Scientifique | | Jacques Gautrais<br>Audrey Dussutour |

The funders had no role in study design, data collection and interpretation, or the decision to submit the work for publication.

### Author contributions

Laure-Anne Poissonnier, Conceptualization, Data curation, Formal analysis, Investigation, Methodology, Writing—review and editing; Sebastien Motsch, Data curation, Formal analysis, Validation, Visualization, Writing—original draft, Writing—review and editing; Jacques Gautrais, Data curation, Software, Formal analysis, Visualization; Camille Buhl, Validation, Writing—review and editing; Audrey Dussutour, Conceptualization, Data curation, Formal analysis, Supervision, Funding acquisition, Methodology, Writing—original draft, Project administration, Writing—review and editing

### Author ORCIDs

Jacques Gautrais http://orcid.org/0000-0002-7002-9920
Audrey Dussutour https://orcid.org/0000-0002-1377-3550

### Decision letter and Author response

Decision letter https://doi.org/10.7554/eLife.48945.023
Author response https://doi.org/10.7554/eLife.48945.024

## Additional files

### Supplementary files

• Supplementary file 1. -Parameter estimations for the four functions. (**Table S1**) Parameter estimations for the four functions with weight: To compensate for the discrepancy in the distribution of data points, we adjust the non-linear fitting by giving more weights for points under-represented (i. e.. increase weight $\omega_i$ for large $k_i$). More precisely, we compute the number $N_i$ of data points for a given density value $k_i$: $N_i = \#\{(k_n, q_n) \text{ with } k_n = k_i\}$. The weight is then given by $\omega_i$1/$N_i$. Using these weights corresponds to performing a fitting $\{k_i, q_i\}$ with $q_i$ the average flow at $k = k_i$. Parameter estimation with uniform weight is performed in Table S2. AW: Akaike weights, RSE: relative standard error. (**Table S2**) Parameter estimations for the four functions without weight. Unlike the Table S1, the parameter estimations use a non-linear regression using all points with the same weight $\omega = 1$.
DOI: https://doi.org/10.7554/eLife.48945.018

• Transparent reporting form
DOI: https://doi.org/10.7554/eLife.48945.019

### Data availability

All data are available from the Dryad Digital Repository: https://doi.org/10.5061/dryad.8q58jg3.

The following dataset was generated:

| Author(s) | Year | Dataset title | Dataset URL | Database and Identifier |
|---|---|---|---|---|
| Poissonnier L-A, Motsch S, Gautrais J, Buhl C, Dussutour A | 2019 | Data from: Experimental investigation of ant traffic under crowded conditions | https://doi.org/10.5061/dryad.8q58jg3 | Dryad Digital Repository, 10.5061/dryad.8q58jg3 |

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
