## [Decision Letter]

Thank you for submitting your article "Still flowing, experimental investigation of ant traffic under crowded conditions" for consideration by *eLife*. Your article has been reviewed by three peer reviewers, one of whom is a member of our Board of Reviewing Editors, and the evaluation has been overseen by Andrew King as the Senior Editor. The following individual involved in review of your submission has agreed to reveal their identity: Daniel Goldman (Reviewer #3).

The reviewers have discussed the reviews with one another and the Reviewing Editor has drafted this decision to help you prepare a revised submission.

This is a nice manuscript looking at how crowding affects ant traffic, based on experimental results for self-organization of traffic in foraging Argentine ant colonies. In this study, the authors collected video recordings of the bidirectional movement of ants on a bridge connecting the nest to a foraging area. Varying the colony size and the bridge width, they collected data on speed, flow, and contacts between ants for a large range of densities. On the basis of these data, the authors proposed a model to explain ant traffic, with good agreement between the model and the data.

Essential revisions:

1) One of the reviewers found that part of the presentation of the model is not sufficiently clear and had the following comments.

* The parameters of Equation 2 need further explanation. It is not clear which type of mechanism is intended to be included with the parameter α and what effects α has on the model. In general, the second term of Equation 2 looks over complicated and some further justification for these parameters would be helpful (as for example Equation 2 could be reduced to Exp[-k*0.08]*2.1).

* "We observed a small decay of the speed *v* up to *k* = 5 followed by a more drastic decay (Figure 6A)". I do not agree with this statement, as it actually seems that the effect of density is lower for large *k* (than for small *k*).

* "Ants might have actively avoided meeting new nestmate as the bridge became crowded". How does this relate to the decay in speed? Active ant avoidance could be a cost that requires moving at a slower speed. Could you comment on this?

* Figure 3 is very unclear. The authors should use the parameters and metrics settings consistently, e.g., density is named *k* and not *d*. These letters should be also included on the plot axes and legend. The vertical axis quantity is not clear and the caption should be rephrased to help the reader to understand the figure.

* It seems that Figure 4A is also a two-phase function (similar to Figure 2A) for *k* > 10.

* What do the white dots in Figure 4 represent? Average and std. deviation? Please include this information in the caption.

* Figure 5B – how has the reported quantity been computed? Please explain.

* Caption of Figure 5B – it is not clear what this means: "*v_f_
*= *L/(T_0_*) i.e. *L/(T-C* Δ*T*)".

* I expected that the data points of Figure 2A and 6B would be the same but they look different. Why?

2) The reviewers believe the paper is missing citations to relevant work which should be commented on (and compared with) on bidirectional flow in ants. For example, take a look at Gravish et al., 2015, for an analogous study in which fire ants trafficked (foraged) in confined (tunnel tubes, models of subterranean foraging networks) environments and a glass-like picture was proposed. Look in the SI for fundamental diagram plots. A major point in that paper is similar to that in the paper under consideration, namely how ants can modulate interaction times (antennation) to avoid clogging. For that matter, many of the conclusions regarding ants in their reference (Aguilar et al., 2018) also discuss how bidirectional trafficking ants can mitigate clogging and maintain their density at the peak of the fundamental diagram (and help avoid dropping of flow rate). The authors should also review Parisi, Soria and Josens, 2014, in which ants must escape from exits and do so without clogging and dropping flow.

3) The reviewers do not think your model should be referred to as a two-phase "model" – the equations for q(v) should be thought of as fitting functions, while the CA "models" in some of the papers the authors cite (and those they should, see above) are models in that a system or scheme is presented and then integrated, calculated, etc. to make predictions. This point should be addressed.

---

## [Author Response]

Essential revisions:1) One of the reviewers found that part of the presentation of the model is not sufficiently clear and had the following comments.* The parameters of Equation 2 need further explanation. It is not clear which type of mechanism is intended to be included with the parameter α and what effects α has on the model. In general, the second term of Equation 2 looks over complicated and some further justification for these parameters would be helpful (as for example Equation 2 could be reduced to Exp[-k*0.08]*2.1).

The second part of the equation “α+β⋅k⋅exp(-γ⋅k)” is used to model the change of the free flow velocity *v_f_* with respect to the density (Figure 5B). This effect is rather subtle since it is not monotone (*v_f_* first increases with k and then decreases). This is one of the reasons why we cannot simply use an exponential function. Without the parameter α, the free velocity *v_f_* would converge exponentially fast to zero with k leading to an Underwood type function. The resulting function would not exhibit a plateau of the flow q at large densities (see Author response image 1).

* "We observed a small decay of the speed v up to k = 5 followed by a more drastic decay (Figure 6A)". I do not agree with this statement, as it actually seems that the effect of density is lower for large k (than for small k).

We changed the sentence to tone down our statement, but if we focus on experimental data (white dot on the figure) speed did decrease slightly more rapidly when *k* > 5 (when *k* = 1 ant.cm^-2^*v* = 1.75cm.s-1, when *k* = 5 ant.cm^-2^*v* = 1.5sec.cm^-1^ (difference 0.25) and when *k* = 9 ant.cm^-2^
*v* = 1 sec.cm^-1^ (difference 0.5)).

* "Ants might have actively avoided meeting new nestmate as the bridge became crowded". How does this relate to the decay in speed? Active ant avoidance could be a cost that requires moving at a slower speed. Could you comment on this?

We agree with the reviewers that active avoidance could be as time consuming as collisions. Yet, after a collision, an ant still need to get around the ant it contacted so directly avoiding a collision might allow the ant to save time. However, as we cannot conclude on this matter we removed this sentence.

* Figure 3 is very unclear. The authors should use the parameters and metrics settings consistently, e.g., density is named k and not d. These letters should be also included on the plot axes and legend. The vertical axis quantity is not clear and the caption should be rephrased to help the reader to understand the figure.

We agree and apologise for the confusion. We changed the legend as suggested by the reviewers and rephrased the caption to better help the reader. We also modified the figure and hope that it is now easier to read.

* It seems that Figure 4A is also a two-phase function (similar to Figure 2A) for k > 10.

Indeed, the data (‘circle dot’) suggests another slope when *k*>10. However, the number of data points is scarce for *k*>10, it represents only 8% of the data. Nonetheless, we did perform a non-linear regression with a two-phase function (see Author response image 2), but the regression leads to a change of slopes at k≈2.8 (ant not at *k*=10) and a minor change of slopes. Since using a two-phase function adds two more parameters to the function (with unclear biological justifications), we choose to use a simpler linear model.

**Author response image 2. respfig2:** 

* What do the white dots in Figure 4 represent? Average and std. deviation? Please include this information in the caption.

It represent average and standard deviation. The information is now included in the legend.

* Figure 5B – how has the reported quantity been computed? Please explain.

We rephrased the legend to better explain the quantity *v_f_*. The free flow speed is the (average) velocity of the ants on the bridge without contacts. We removed the time spent due to contacts by reducing the measured traveling T by *C*⋅Δ*T* where Δ*T* represents the average time spent per contact (Δ*T* = .24s from figure 5A). The free flow speed is then estimated as: *v_f_* = *L/(T-C*⋅Δ*T*), i.e. distance traveled divided by time spent without contacts.

* Caption of Figure 5B – it is not clear what this means: "v_f_ = L/(T_0_) i.e. L/(T-C ΔT)"

We clarified the caption (see previous answer).

* I expected that the data points of Figure 2A and 6B would be the same but they look different. Why?

Individual behavior were measured on a subset of experiment thus Figure 6B show the experimental observations of the sub-sample dataset (N=7900 observations) used for the estimation while Figure 2A show the full dataset (N = 612000). We now made that clearer in the caption of figure 6B to avoid any confusion. We added “Experimental data correspond to the observations of a sub-sample dataset (7900 observations) from the full data set (612000 observations; Figure 2A) used to quantify the individual behavior.”

2) The reviewers believe the paper is missing citations to relevant work which should be commented on (and compared with) on bidirectional flow in ants. For example, take a look at Gravish 2015, for an analogous study in which fire ants trafficked (foraged) in confined (tunnel tubes, models of subterranean foraging networks) environments and a glass-like picture was proposed. Look in the SI for fundamental diagram plots. A major point in that paper is similar to that in the paper under consideration, namely how ants can modulate interaction times (antennation) to avoid clogging. For that matter, many of the conclusions regarding ants in their reference (Aguilar et al., 2018) also discuss how bidirectional trafficking ants can mitigate clogging and maintain their density at the peak of the fundamental diagram (and help avoid dropping of flow rate). The authors should also review Parisi, Soria and Josens, 2014, in which ants must escape from exits and do so without clogging and dropping flow.

We have now given Aguilar et al., 2018, a more prominent place and we are grateful for the pointer to the interesting studies from Gravish et al., 2015 and Parisi, Soria and Josens, 2014, which we have now cited and discussed.

3) The reviewers do not think your model should be referred to as a two-phase "model" – the equations for q(v) should be thought of as fitting functions, while the CA "models" in some of the papers the authors cite (and those they should, see above) are models in that a system or scheme is presented and then integrated, calculated, etc. to make predictions. This point should be addressed.

We agree with the reviewers that we build fitting functions. In our paper, we used the term “model” as it used for example in statistics: general linear model, non-linear regression models etc. We changed the text accordingly and replace model by functions or when using the term model to describe statistics we used “statistical model”.